# Transcriptional Stress Induces the Generation of DoGs in Cancer Cells

**DOI:** 10.3390/ncrna10010005

**Published:** 2024-01-10

**Authors:** Francisco Rios, Maritere Uriostegui-Arcos, Mario Zurita

**Affiliations:** Departamento de Genética del Desarrollo y Fisiología Molecular, Instituto de Biotecnología, Universidad Nacional Autónoma de México, Av. Universidad 2001, Cuernavaca Morelos 62250, Mexicomtua@bu.edu (M.U.-A.)

**Keywords:** DoGs, transcription, stress, TFIIH, triptolide, THZ1, cancer

## Abstract

A characteristic of the cellular response to stress is the production of RNAs generated from a readthrough transcription of genes, called downstream-of-gene-(DoG)-containing transcripts. Additionally, transcription inhibitor drugs are candidates for fighting cancer. In this work, we report the results of a bioinformatic analysis showing that one of the responses to transcription inhibition is the generation of DoGs in cancer cells. Although some genes that form DoGs were shared between the two cancer lines, there did not appear to be a functional correlation between them. However, our findings show that DoGs are generated as part of the cellular response to transcription inhibition like other types of cellular stress, suggesting that they may be part of the defense against transcriptional stress.

## 1. Introduction

In recent years, different drugs (both traditional and new-generation) that affect components of the basal transcription machinery—particularly those mediated by RNA polymerase II (RNA pol II)—have been proposed for use against cancer [1,2,3]. In fact, some of these substances have been tested in clinical trials [4,5,6,7].

The targets of these substances are mainly components of the preinitiation complex (PIC) or factors that are associated with the PIC either during transcription initiation or during transcription elongation [8,9]. Among these drugs are substances that preferentially kill cancer cells—for example, inhibitors of the RNA pol II itself, such as actinomycin D and α-amanitin [10]; inhibitors of the transcription and DNA repair complex TFIIH, such as triptolide (TPL) and THZ1; inhibitors of the PTEFb complex such as flavopiridol; and others that affect transcription [11,12,13,14]. Additionally, compounds have been developed that affect the functions of the elongation machinery—for example, JQ1, which inhibits BRD4 and has a strong antiproliferative effect on different cancer cells [1,15].

Recently, we reported that some genes are overexpressed when breast cancer cells are treated with TPL and THZ1, despite the transcription of a large number of genes being inhibited [16]. In addition, the inhibition of TFIIH, as well as the inhibition of transcription initiation and elongation in other types of cancer cells using other substances, also induces the overexpression of a significant number of genes, some of which are shared among the responses to these substances [10]. It is obvious that, since cancer cells are exposed to a stress condition that inhibits transcription, they respond by overexpressing some genes, and we have named this the transcriptional stress response (TSR) [16]. Interestingly, the reduction in the expression of some of these transcriptional stress response genes enhances the effect of TPL [16]. Therefore, TSR is another type of cellular stress response that includes the overexpression of specific genes.

A characteristic that has recently been noted in cellular responses to different types of stress, such as osmotic stress and heat shock, as well as in some tumor cells and during viral infections, is the production of long noncoding RNAs (lncRNAs) generated from a readthrough transcription of genes transcribed by RNA pol II; in other words, these transcripts are not processed at the site where these events normally occur downstream of the polyadenylation site (PAS) [17,18,19,20,21,22,23,24]. These lncRNAs have been called downstream-of-gene (DoG)-containing transcripts. In general, the transcription of these DoGs is initiated in the promoter of a gene transcribed by RNA pol II. A DoG transcript has a minimum length of 5 kb and begins at the transcription termination site in the 3′ of the gene, although the length of this type of RNA can be different among genes, cellular stresses, and even cell types. DoGs are retained in the nucleus, and it is likely that they remain associated with chromatin [20,25,26]. It has been documented that alterations in the transcription termination machinery, such as the cleavage and polyadenylation (CPA) complex or the XRN2 exonuclease, can result in the generation of DoGs [26,27]. Likewise, it has been reported that hyperosmotic stress causes a disruption of the Int11 subunit of the Integrator complex, which is an endonuclease that participates in terminating the transcription of some genes, triggering the production of DoGs [22]. Currently, it is unknown whether these lncRNAs have a role in the stress response or are simply the product of transcriptional dysregulation, which intriguingly preferentially affects some but not all the genes in the genome.

With this information as a basis, we formulated the question of whether transcriptional stress, similar to other types of stress, also results in the generation of DoGs. To answer this question, we analyzed our transcriptome data from breast cancer cells treated with TPL [16], which inhibits transcription initiation, as well as public transcriptome data from other types of cancer cells treated with TPL and THZ1, which also inhibit transcription. In general, we found that TSR also induces the generation of DoGs, and an important proportion of these DoGs are shared between cell types. This confirms that the inhibition of transcription induces a response similar to that induced by other types of stress and that it may be part of a mechanism to protect cancer cells against the effects of substances that inhibit transcription.

## 2. Results

### 2.1. Inhibition of Transcription by TPL Induces the Formation of DoGs in Breast Cancer Cells

TPL affects the initiation of transcription mediated by RNA pol II, inhibiting the ATPase activity of the XPB subunit of TFIIH, which is essential for the formation of the open complex, and induces the dissociation of the XPB-p52-p8 module from TFIIH [16]. As mentioned above, although TPL inhibits transcription, some genes are overexpressed [16]. While analyzing the transcripts by a visualization of RNA-seq data obtained during the TSR in the MCF10-Er-Src transformed cell line, we found that some genes contained reads extending for several kb beyond the 3′ end. Based on this observation, we decided to analyze whether the TSR, similar to other types of cellular stress, induces the formation of DoGs. To this end, we analyzed the RNA-seq data of MCF10A-Er-Src cells with and without TPL treatment using ARTDeco [28], which have been used by several groups to identify DoGs from RNA-seq data. The minimum length of DoGs is generally considered to be approximately 5 kb; however, this is an arbitrary consideration. We decided to consider a minimum length of 4 kb of DoGs, since, when comparing many genes in the presence or absence of transcriptional stress, it is obvious that many genes have a clear readthrough extension of approximately 4 kb when transcription is inhibited. For example, in the case of breast cancer cells, we detected 374 DoGs with extensions between 4000 and 5000 nucleotides (as show in Appendix A).

The RNA-seq data from the breast cancer cells were single-end and non-stranded, and this might generate false positives using ARTDeco, as the authors mention in the program’s documentation [28]. Therefore, to use the RNA-seq data from non-stranded breast cancer cells and ARTDeco, we took advantage of RNA pol II ChIP-seq data from the same cells under the same conditions. It is known that, in most genes that are transcriptionally active, RNA pol II pauses after synthesizing between 20 and 120 nt. Taking this data into account, we filtered the genes that produce DoGs by analyzing the presence of the RNA pol II peak in the 5′ region of the gene. In addition, another filter was applied by excluding those DoGs with RNA pol II peaks that map in their DoG region (see Section 4). Thus, we were able to use the DoGs obtained by ARTDeco with confidence only from those genes that were transcriptionally active.

An example of a gene that generates DoGs in response to transcription inhibition in MCF10A-Er-Src cells is shown in Figure 1A. ARTDeco identified 789 DoGs in TPL-treated cells (Figure 1B; Appendix A). Although 326 DoGs were identified in untreated cells, the number was lower than in the cells treated with TPL (Figure 1B). We observed that 183 of the DoGs identified in TPL-treated cells were only shared with cells incubated with DMSO (Figure 1C). To know the expression levels of the DoGs for each condition, we performed a statistical analysis using ARTDeco ‘diff_exp_dogs’ mode (that use DESeq2), obtaining the log2 fold change and *p*-values from DoGs comparing the expression in TPL-treated cells to the expression in untreated cells. Of the 932 DoGs identified in cells treated with TPL and with DMSO, 651 had a *p*-value ≤ 0.05. This analysis is represented in a heat map (Figure 1D) that shows two areas: one from untreated cells (blue area) and another one from TPL-treated cells (red area); this indicates that the levels of downstream transcripts identified by ARTDeco in the cells treated with TPL were significantly higher than in the control cells (Figure 1D; Appendix A). Similar criteria were used for the analysis of pancreatic cells (see ahead).

In response to transcriptional stress, some genes were overexpressed. To determine if there was a correlation between genes that generated DoGs and genes that were over transcribed in response to TPL, we performed a scatter plot analysis of the expression of each DoG vs. the expression of the corresponding gene (using the ARTDeco ‘diff_exp_read_in’ mode to obtain the log_2_ fold change of DoG-producing genes) and calculated the Pearson’s correlation coefficient with the gene’s transcripts that had a *p*-value ≤ 0.05 both in the DoG region and in the body of the gene (n = 369) (Figure 1E).

Interestingly, there was no correlation between the expression levels of genes that were overexpressed in response to transcriptional stress and those from which DoGs were generated (Figure 1E) [16]. This indicates that the generation of DoGs is not due to the overexpression of a gene, at least not in all cases. Taken together, the data analyzed in this section demonstrate that DoGs are generated as a part of the response to transcriptional stress in breast cancer cells.

### 2.2. Transcriptional Stress Also Induces the Generation of DoGs in Pancreatic Cancer Cells

To investigate whether DoGs were generated in other types of cancer cells treated with TPL to inhibit transcription, we analyzed TPL-treated cells derived from pancreatic cancer in public RNA-seq data [29]. Similar to our observations in MCF10A-Er-Src cells, we detected the presence of DoGs in these pancreatic cells following TPL treatment. An example of a gene generating a DoG is shown in Figure 2A, which corresponds to the same gene depicted in Figure 1A, demonstrating a similar DoG formation in pancreatic cells.

Intriguingly, as in MCF10A-Er-Src cells, untreated pancreatic cancer cells also displayed DoGs formation (Figure 2B). However, treatment with TPL induced the generation of DoGs from a larger number of genes in this cell line (Figure 2B; Appendix A). When comparing the DoGs in DMSO-treated and TPL-treated pancreatic cancer cells, we found that 1406 DoGs were shared. And when those DoGs were excluded from the TPL-treated cells, 2152 were found to be generated in response to transcriptional stress. This number was higher than that of DoGs in the MCF10A-Er-Src transcriptome and was due to differences in the sequencing depth in each experiment. Although the number of DoGs detected by ARTDeco was different between the two cell lines, out of the 606 found in breast cancer cells, 224 were also present in pancreatic cells (Figure 2C; Appendix A).

Similarly, the number of reads for the transcripts identified as DoGs generated by the TPL treatment in the pancreatic cells was quantified, and the corresponding values were determined and compared in a heatmap with the cells treated with DMSO, showing a clear increase in RNAs downstream of the identified genes (Figure 2D, Appendix A).

Additionally, as with breast cancer cells, there did not appear to be a correlation between genes with transcriptional upregulation in response to transcription inhibition and those that generated DoGs (Figure 2E). In summary, the analysis presented in this section shows that transcription inhibition not only results in the generation of DoGs in the MCF10A-Er-Src cell line but also in other cancer cell types.

### 2.3. Treatment of Pancreatic Cancer Cells with THZ1 Also Induces the Formation of DoGs

THZ1 is an inhibitor of the kinase activity of the Cdk7 subunit of TFIIH. Therefore, it is also an inhibitor of transcription mediated by RNA pol II and is a substance that is under investigation for the treatment of cancer [8,30]. Therefore, we decided to explore whether Cdk7 inhibition in cancer cells also induced the formation of DoGs. Again, we used public RNA-seq data from a panel of pancreatic cancer cells [29] and ARTDeco. As expected, we found an increase in DoGs in the cells treated with THZ1. An example of a gene with its DoG is shown in Figure 3A. As mentioned before, DoGs were also identified in untreated cells. However, we observed that the number of DoGs found by ARTDeco in THZ1-treated cells (3018 DoGs) was much larger than those in untreated cells (Figure 3B; Appendix A). Again, as we did above, the DoGs found in THZ1 were compared to those from untreated cells, and 1049 DoGs were observed to be shared; leaving 1969 DoGs that were exclusively expressed in pancreatic cancer cells treated with THZ1.

From this analysis, we found that a large number of DoGs generated in response to TPL and THZ1 treatment were shared between breast cancer and pancreatic cancer cells (Figure 3C; Appendix A). Intriguingly, even though both substances affected two different activities of TFIIH, there were DoGs that were unique for each treatment, even though the experiments were performed in the same cell type. However, among the genes detected in breast cancer cells in response to TPL, 92 were also detected in pancreatic cancer cells treated with THZ1, and 67 were shared between the pancreatic cells treated with THZ1 and TPL, as well as the breast cancer cells treated with TPL (Figure 3C). This suggests that certain genes, when the activity of TFIIH is inhibited, are more likely to exhibit improper transcription termination in different cancer cells.

Once again, the difference in the expression levels of DoGs was obtained, and a heatmap showing the log_2_ fold change was plotted (Figure 3D). The heat map shows that DoGs exclusively found in THZ1-treated cells have a significantly higher expression in those cells than in untreated cells.

Taking into account the expression levels of the DoG-producing genes and DoGs from THZ1- and DMSO-treated cells (with *p*-value ≤ 0.05 in both the gene and the DoG region), we elaborated a representational scatter plot (Figure 3E), demonstrating again that the expression level from the DoG-producing gene did not correlate with the expression level of the DoG.

### 2.4. Most of the DoGs Generated in Response to Transcriptional Stress Do Not Overlap with the DoGs Generated in Response to Osmotic Stress

The observation that some DoGs are generated in both breast and pancreatic cancer cells in response to transcriptional stress raises the question of whether these DoGs are also generated under other types of stress. To explore this, we compared the DoGs recently reported to be generated in response to hyperosmotic stress in HEK293T cells [22] and those observed in breast cancer and pancreatic cells after TPL and THZ1 treatment. Out of the 606 DoGs generated in breast cancer cells in response to transcriptional stress, only 41 overlapped with those reported in cells subjected to osmotic stress (Figure 4A). Similarly, in pancreatic tumor cells treated with TPL and THZ1, only a subset of the DoGs was shared with those induced by hyperosmotic stress (Figure 4C,E). This indicates that different types of stress can induce the formation of DoGs from different genes, although this phenomenon may also depend on the cell type (see Section 3).

We also sought to determine whether particular types of genes with specific functions were more likely to generate DoGs. To this end, we performed an ontological analysis of the DoGs generated in the breast cancer cell line and the pancreatic cancer cells (Figure 4B,D,F). While the ontological analysis revealed a wide effect on different molecular functions of the genes that produce DoGs in the TSR, it is intriguing that many of these genes were related to processes that involved the synthesis of RNA (Figure 4B,D,F). However, at this point, it was not possible to correlate the genes that generated DoGs in the TSR with a specific function. In addition, we performed a GO analysis on the 668 DoGs shared by pancreatic cells treated with TPL and THZ1, as well as on the 67 DoGs found in the three conditions analyzed (Appendix A). Although the 668 genes shared by cancer pancreatic cells treated with the two drugs tended to be related to molecular functions associated with gene expression, the 67 DoGs found in the three conditions did not show a clear correlation with a specific cellular function.

## 3. Discussion

The response to different types of stress is fundamental for the survival of cells when they are subjected to different types of insults. On the other hand, the stress response can also determine whether it is preferable for a cell to die when the damage is very severe. In general, any insult that generates stress in a cell causes the inactivation and overexpression of genes. A characteristic observed under oxidative stress, heat shock, and viral infections is the generation of DoGs [18,19,20,21,22,23,31]. We recently described how stress due to inhibition of the initiation of transcription mediated by RNA pol II causes the overexpression of some genes [16]. With the analysis presented here, we show that the TSR in breast cancer cells also leads to the generation of DoGs. This reinforces the fact that, after the inhibition of transcription initiation using substances that inhibit the enzymatic activities of the TFIIH complex components, the cellular response is similar to that after other types of insults.

Intriguingly, we did not find a correlation between genes that are overexpressed in response to transcription inhibition and those that generate DoGs. This suggests that the mechanism by which some transcripts are extended in response to stress does not depend on an increase in transcription. We also found that the DoGs generated in response to TPL treatment are not exclusive to breast cancer cells but are also generated in cells derived from pancreatic cancer. This suggests that there are genes that, either due to their function or due to the characteristics of their location in the genome, are more prone to DoG generation. However, by ontological analysis, we found that there does not seem to be a correlation among the functions of these genes, and when these genes are compared with genes from which DoGs are generated in response to hyperosmotic stress, there is not a strong correlation. This is a pattern that other groups have observed during investigations of different types of stress in different cell types [20,22,26]. In addition, we found that an important number of DoGs generated by TPL and THZ1 in pancreatic cancer cells are unique for each treatment, even that both substances are inhibitors of transcription. These could be related to the fact that TPL is a more severe drug than THZ1, and the generation of some DoGs is therefore different. Also, side effects cannot be discarded; however, more research is needed to answer these discrepancies. In addition, in all the experiments analyzed here, DMSO is the vehicle that is used, both with TPL and THZ1. In all cases, DoGs were detected in cells treated with DMSO. Therefore, that DMSO causes some type of stress that can induce the generation of DoGs cannot be ruled out.

A common feature when transcription is affected is the degradation of the RNA pol II large subunit, as is the case when TFIIH is affected [16,32]. Thus, other substances that affect PIC components may also generate DoGs. Therefore, it will be relevant in a future analysis to determine if the different inhibitors of the RNA pol II also generate DoGs. It is possible that the inhibition of transcription has an impact effect in the factors that participate in transcription. Recent studies of cells exposed to osmotic stress have shown that the Integrator catalytic subunit Int11, which is required for the correct termination of many transcripts, hyperosmotic stress induce its ubiquitination and, in consequence, its depletion, resulting in DoG generation [22]. It would be interesting in future studies to determine if the Integrator complex is also affected during the TSR and if it is also the cause of DoG formation. On the other hand, the question as to whether the mechanism in the generation of DoGs is the same in different cellular stress conditions or not remains open, as well as if the integrator is involved in all cases or if other factors are also involved.

The transcripts that form DoGs remain sequestered in the nucleus and are therefore not translated [20,26]. Therefore, it has been proposed that their retention in the nucleus may be part of a stress protection mechanism to maintain chromatin integrity [24]. If so, the observation that RNA pol II extends the transcription elongation of a gene to regions that are not normally transcribed implies that the chromatin in those sites must be more open. Whether this renders the chromatin more protected against or more sensitive to damage is undetermined. On the other hand, the fact that mRNA with a DoG remains in the nucleus and cannot therefore be translated could be an alternative mechanism to inhibit the production of proteins when the cell is under stress. The fact that DoGs are generated during viral infections supports this hypothesis [25]. However, a limitation of the analysis presented in this work is that there is only information on the use of drugs that inhibit transcription in cancer cells. Therefore, it is highly probable that, in healthy cells, DoGs are also generated in response to this type of insult. It will be interesting to determine if this is something exclusive to the response of cancer cells to transcriptional stress or if it is a general phenomenon.

In conclusion, the inhibition of transcription generates a typical stress response in cells in which not only is the overexpression of certain genes induced but DoGs are also generated, as under other types of cellular stress. These two types of responses are particularly important for the treatment of cancer cells with transcription inhibitors, since their effects may be involved in the generation of cells resistant to treatment.

## 4. Materials and Methods

### 4.1. Data Collection

FastQC-v0.11.7 was used to assess the quality of the sequence data. RNA-seq reads were aligned to the hg19 reference genome using Bowtie2 v. 2.3.4.3 with default parameters. Samtools v. 1.9 was utilized to generate BAM files for each experiment and replicate.

### 4.2. DoGs Identification

To identify DoGs, BAM files from breast and pancreatic cancer cells were processed using ARTDeco ‘get_dogs’ mode. A GTF file from UCSC (hg19 from Ensembl) was used with default settings, including a minimum DoG length of 4 kb, a DoG window size of 500 bp, and a minimum FPKM of 0.2 for DoG discovery.

### 4.3. Filtering DoGs with ChIP-seq Data

DoGs BED files generated by ARTDeco for MCF10A-Er-Src cells were used. RNAP II ChIP-seq BED files containing promoter peak coordinates for both DMSO and TPL conditions were obtained from GEO accession GSE135256 (Appendix A).

Two filters were applied:

Filter 1: Checked for peaks that mapped inside DoGs. If a DoG covered a downstream gene with an RNAP II peak, it was considered a potential transcription of the downstream gene and discarded.

Filter 2: Looked for DoG-producing genes with an RNAP II peak in their promoters to validate their transcription status.

Both filters were implemented using Python scripts available on the GitHub project page as Filter_1_with_ChIPseq.py and Filter_2_with_ChIPseq.py.

### 4.4. DoGs Count and Comparison

The number of DoGs for each condition, replicate, and cell line was determined by counting the lines in each BED file generated by ARTDeco.

To identify common DoGs between replicates, conditions, or cell lines, a Python script named Common_DoGs.py was used. This script searched for similar Ensembl IDs assigned to DoGs in two different BED files. To merge two or more DoGs files, the Union_DoGs_annotation command from DoGFinder was employed [33]. Bar graphs and Venn diagrams were created using R for visualization.

### 4.5. Statistical Analysis

ARTDeco ‘diff_exp_read_in’ and ‘diff_exp_dogs’ modes were used to obtain the log2 fold change and *p*-values from DoG-producing genes and DoGs, respectively. With the output of those files and the DoGs files mentioned above, heatmaps and scatter plots were generated with Python scripts (the Pearson’s correlation coefficient showed in the scatter plots was calculated using the SciPy library) available on the GitHub project page as Heatmap_log2fc.py and Scatterplot.py.

Boxplots were created using the modified DoG files, including DoG length information, and executed with the Boxplot.py code from the GitHub project page.

### 4.6. Comparison of Genes

Names of DoG-producing genes from osmotic stress were acquired from Appendix A [22]. Ensembl IDs for DoGs from TPL and THZ1 treatments in both cell lines were extracted and converted using DAVID Bioinformatics Resources to official gene symbols [34,35]. The Stress_comparison.py code was utilized to identify common genes between DoG-producing genes from TPL or THZ1 and those from osmotic stress.

### 4.7. Gene Ontology Analysis

Gene ontology analysis of biological processes was conducted using EnrichR [36,37].

## Figures and Tables

**Figure 1 ncrna-10-00005-f001:**
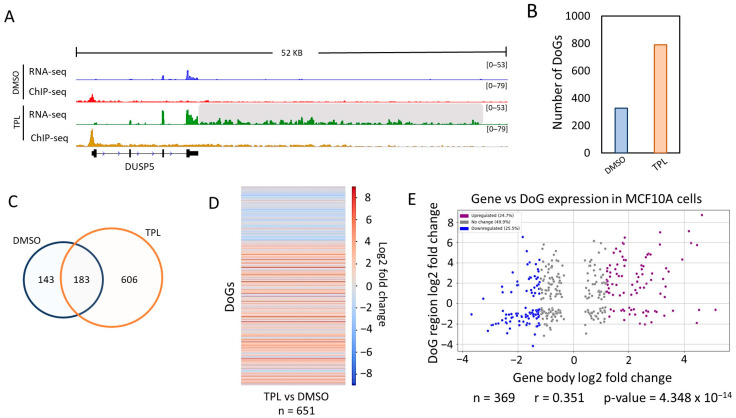
Inhibition of the XPB subunit of TFIIH by TPL induces the formation of DoGs in breast cancer cells. (**A**) Browser image of the RNA-seq and ChIP-seq profile of RNA pol II from MCF10A-Er-Src cells treated with DMSO or TPL. An example of a DoG-producing gene (DUSP5) identified with ARTDeco is presented, and the DoG region is delineated by a gray shadow (log_2_FC = 5.11 and *p*-value = 9.65 × 10^−71^); (**B**) Quantification of DoGs in MCF10A-Er-Src cells with DMSO and TPL; (**C**) Venn diagram showing that 183 DoG-producing genes are shared between breast cancer cells treated with TPL and DMSO; (**D**) Heatmap that exhibits the log_2_FC for each DoG from 651 DoGs found with TPL and DMSO with *p*-value ≤ 0.05; (**E**) Scatter plot displaying the log_2_FC for the transcription levels of the DoG-producing genes (*p*-value ≤ 0.05) on the *x*-axis and the log_2_FC for the corresponding DoGs (*p*-value ≤ 0.05) on the *y*-axis. DoG-producing genes were classified as ‘Upregulated’ (24.7%; purple dots) with log_2_FC ≥ 1.2, ‘Downregulated’ (25.5%; blue dots) with log_2_FC ≤ −1.2, and ‘No change’ (49.9%; gray dots) with log_2_FC between 1.2 and −1.2. Percentages for each category are displayed in the legend box. Pearson’s correlation coefficient and *p*-value were calculated, and both are shown below the graph.

**Figure 2 ncrna-10-00005-f002:**
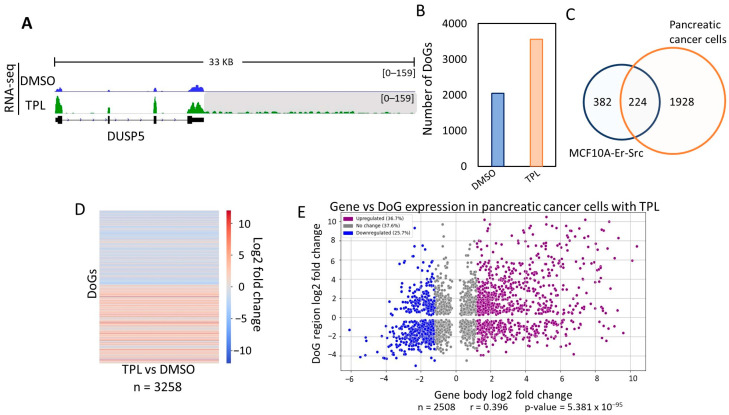
Inhibition of transcription by TPL in pancreatic cancer cells induces the formation of DoGs. (**A**) Browser image of RNA-seq from pancreatic cancer cells treated with DMSO or TPL. The same example of a DoG-producing gene (DUSP5) also found in breast cancer cells in Figure 1A is shown, and the DoG region is indicated by a gray shadow (log_2_FC = 3.71 and *p*-value = 1.52 × 10^−27^); (**B**) Number of genes in pancreatic cancer cells in which there is the formation of DoGs in cells treated with DMSO and in response to the inhibition of transcription by TPL; (**C**) Venn diagram showing that, of the 606 genes that generate DoGs in breast cancer cells specific to the response to TPL, about 37% (224 DoGs) were also found in pancreatic cancer cells in response to TPL; (**D**) Heatmap that exhibits the log_2_FC for each DoG from the 3258 DoGs found with TPL and DMSO in pancreatic cancer cells with *p*-value ≤ 0.05; (**E**) Scatter plot displaying the log_2_FC for the transcription levels of the DoG-producing genes (*p*-value ≤ 0.05) on the *x*-axis and the log_2_FC for the corresponding DoGs (*p*-value ≤ 0.05) on the *y*-axis. DoG-producing genes were classified as ‘Upregulated’ (36.7%; purple dots) with log_2_FC ≥ 1.2, ‘Downregulated’ (25.7%; blue dots) with log_2_FC ≤ −1.2, and ‘No change’ (37.6%; gray dots) with log_2_FC between 1.2 and −1.2. Percentages for each category are displayed in the legend box. Pearson’s correlation coefficient and *p*-value were calculated, and both are shown below the graph.

**Figure 3 ncrna-10-00005-f003:**
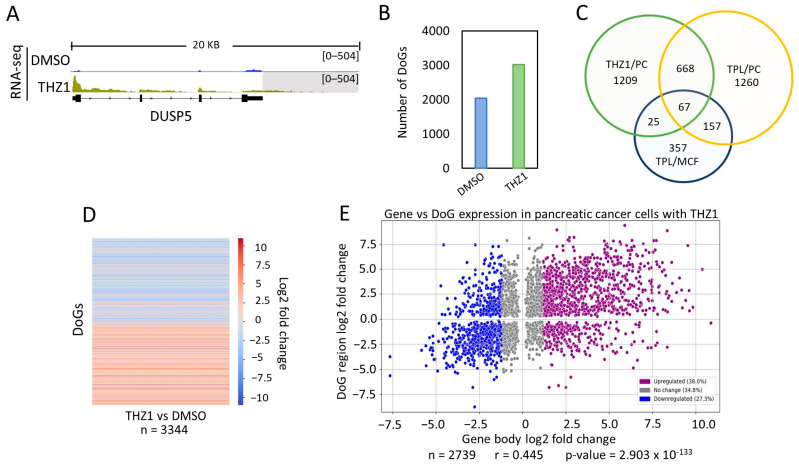
The inhibition of transcription with THZ1 induces the production of DoGs in pancreatic cancer cells. (**A**) Browser image of RNA-seq from pancreatic cancer cells treated with DMSO or THZ1, showing the DUSP5 gene that also undergoes the generation of a DoG in response to THZ1. DoG region is indicated by a gray shadow (log_2_FC = 3.83 and *p*-value = 3.06 × 10^−29^); (**B**) Number of genes that generate DoGs in response to DMSO and THZ1 in pancreatic cancer cells; (**C**) Venn diagram showing the genes that are shared between pancreatic cancer cells treated with TPL and THZ1 and with breast cancer cells treated with TPL; (**D**) Heatmap showing the log_2_FC of downstream readthrough transcripts (*p*-value ≤ 0.05) between pancreatic cancer cells treated with THZ1 and DMSO; (**E**) Scatter plot displaying the log_2_FC for the transcript levels of the DoG-producing genes (*p*-value ≤ 0.05) on the *x*-axis and the log_2_FC for the corresponding DoGs (with *p*-value ≤ 0.05) on the *y*-axis. DoG-producing genes were classified as ‘Upregulated’ (38.0%; purple dots) with log_2_FC ≥ 1.2, ‘Downregulated’ (27.3%; blue dots) with log_2_FC ≤ −1.2, and ‘No change’ (34.8%; gray dots) with log_2_FC between 1.2 and −1.2. Percentages for each category are displayed in the legend box. Pearson’s correlation coefficient and *p*-value were calculated, and both are shown below the graph.

**Figure 4 ncrna-10-00005-f004:**
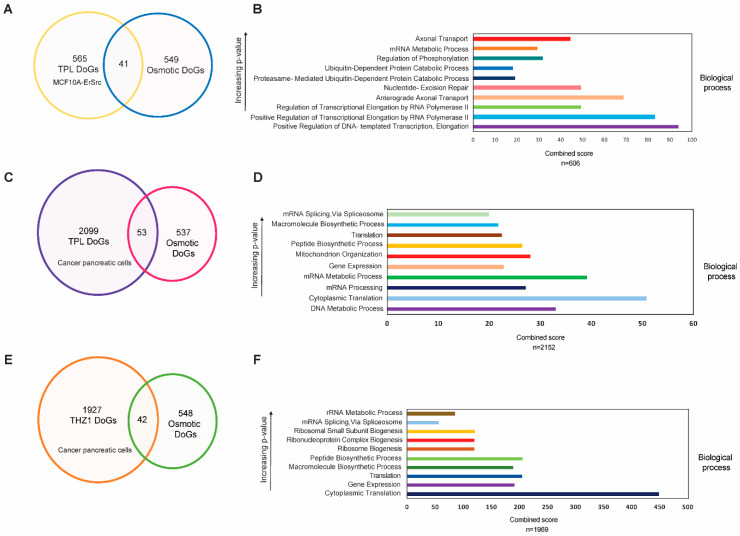
DoGs generated in response to transcriptional stress are mostly different from those generated by osmotic stress and are not related to a specific cell function. (**A**) Venn diagram showing that, of the 606 DoGs generated by transcriptional stress in cancer breast cells, only 41 are also produced by osmotic stress; (**B**) Ontological analysis showing the biological functions of genes in which DoGs are generated by transcriptional stress in the breast cancer cells; (**C**) Venn diagram showing that only 53 DoGs are shared between the cancer pancreas cells generated by TPL treatment and produced by osmotic stress; (**D**) Ontological analysis showing the biological functions of genes in which DoGs are generated by TPL treatment in pancreatic cancer cells; (**E**) Venn diagram showing that only 42 DoGs are shared between the cancer pancreas cells generated by THZ1 treatment and produced by osmotic stress; (**F**) Ontological analysis showing the biological functions of genes in which DoGs are generated by THZ1 treatment in pancreatic cancer cells.

## Data Availability

RNA-seq raw data was downloaded from NCBI using SRA-tools; for breast cancer cells, GEO accession: GSE135256 and RNA-seq_T-1, RNA-seq_T-2, RNA-seq_TT-1, RNA-seq_TT-2 samples were used (GEO accession: GSM3999751, GSM3999752, GSM3999753, GSM3999754); for pancreatic cancer cells, GEO accession: GSE157927 and pancreatic cancer cells—Vehicle (DMSO)-treated RNA-Seq Replicate 1, pancreatic cancer cells—Vehicle (DMSO)-treated RNA-Seq Replicate 2, pancreatic cancer cells—Triptolide-treated RNA-Seq Replicate 1, pancreatic cancer cells—Triptolide-treated RNA-Seq Replicate 2, pancreatic cancer cells—THZ1-treated RNA-Seq Replicate 1, pancreatic cancer cells—THZ1-treated RNA-Seq Replicate 2 samples were used (GEO accession: GSM4781045, GSM4781046, GSM4781047, GSM4781048, GSM4781049, GSM4781050, respectively). GitHub Project page for code and data: https://github.com/PacoRM24/Transcriptional_stress_response_and_readthrough_transcription (accessed on 18 October 2023).

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
