# Peer review of "Transcriptional Stress Induces the Generation of DoGs in Cancer Cells"

_ncrna, 2024, doi:10.3390/ncrna10010005_

Round 1

Reviewer 1 Report

Comments and Suggestions for Authors

In this manuscript Prof. Zurita and coworkers report that inhibition of TFIIH leads to the upregulation of transcript in two cancer cell lines. A fraction of these transcripts show readthrough transcription (DoGs). Notably, DoG-transcripts detected upon TFIIH inhibition in the two cell lines are different, moreover DoGs triggered by osmotic stress in the PC cell lines are different from those identified upon TFIIH inhibitors. This work supports the notion that DoG might be generated by a variety of cellular stress in a context dependent manner.

Major points:

(1)   The author state in the title that DoG are generated in cancer cell upon trx stress, yet they only analyze cancer cells, thus there is no indication that this is a cancer specific response. Since there are plenty of publicly available datasets (mRNA-seq), it should be feasible to verify whether non-cancer cells generate DoG transcripts upon transcriptional stress. In the absence of such data, any reference to cancer cell specificity should be removed from the text and title.

(2)   I did not find the Supplementary figure and material, so I could not evaluate them. In any case, please include tables with the full list of differentially expressed genes and DoGs (fold change, statistics).

Author Response

First, we would like to thank the reviewers for their comments and for the time dedicated to reviewing this manuscript. The suggested comments have significantly improved this work.

Response to the reviewer 1

Reviewer concern 1:

The author state in the title that DoG are generated in cancer cell upon trx stress, yet they only analyze cancer cells, thus there is no indication that this is a cancer specific response. Since there are plenty of publicly available datasets (mRNA-seq), it should be feasible to verify whether non-cancer cells generate DoG transcripts upon transcriptional stress. In the absence of such data, any reference to cancer cell specificity should be removed from the text and title.

Answer:

This is a very interesting point made by the reviewer. To the best of our knowledge, the use of TPL and THZ1 to inhibit TFIIH activities and therefore RNA pol II-mediated transcription has only been tested in cancer cells, and transcriptomic data are not available in all cases. Therefore, we agree with the reviewer that we cannot say that DoGs are not generated in response to transcriptional stress in non-cancerous cells, but nor can we say that they are generated. Drugs that inhibit transcription have only been tested in cancer cells for their potential use in cancer therapies. Since the data that were used in this manuscript are from works focused on seeing the effect of these substances to kill cancer cells, we consider that it is important to maintain the focus of this manuscript on cancer cells, since the generation of DoGs in response to this insult may be part of a defense mechanism against this type of therapy. In the new version, we now detail this point in the discussion by stating the following on Page 9, line 334:

"A limitation of the analysis presented in this work is that there is only information on the use of drugs that inhibit transcription in cancer cells. Therefore, it is highly probable that in healthy cells, DoGs are also generated in response to this type of insult. It will be very interesting to determine if this is something exclusive to the response of cancer cells to transcriptional stress or if it is a general phenomenon."

Reviewer Concern 2:

 I did not find the Supplementary figure and material, so I could not evaluate them. In any case, please include tables with the full list of differentially expressed genes and DoGs (fold change, statistics).

Answer:

We apologize for this inconvenience. We included both the supplementary figures and the tables with the data of all the genes, along with all the statistics, in the first version. We do not understand what happened. In this new version, we are resubmitting the manuscript with all the supplementary information.

Reviewer 2 Report

Comments and Suggestions for Authors

The authors identified certain DoGs (RNAs generated from readthrough transcription of genes) in cancer cell lines, with or without PolII inhibitor and with some stress. Although some DoGs were common, no correlation was found.
The article is linear, with no way out, and easily readable.

However, I have a few minor/major points to make:
- Line 119: dogs should be written as DoGs
- Figure 1, 2, 3: The Log2FC used is 1.2. Usually, 1.5 is used. What would be the results with such a limit?
- DMSO is used as a vehicle and therefore as a control. I understand that this cannot be changed. But I would like to see a paragraph in the discussion stating that DMSO probably also induced DoGs as this is a major stress for the cells.

Author Response

First, we would like to thank the reviewers for their comments and for the time dedicated to reviewing this manuscript. The suggested comments have significantly improved this work.

Answers to the reviewer concerns:

Reviewer Concern 1.

Line 119: dogs should be written as DoGs

Thank you for the observation. This has been corrected in the new version.

Reviewer Concern 2.

Figure 1, 2, 3: The Log2FC used is 1.2. Usually, 1.5 is used. What would be the results with such a limit?

Answer:

This is a very interesting observation by the reviewer. The Log2FC of 1.2 was used in the generation of the scatter plots for the readings of the genes that are overexpressed with TPL and to correlate it with those that are generated DoGs in response to the inhibition of transcription. As the reviewer suggests, we changed the limit to 1.5 of the Log2FC and the result is the same and does not change the conclusions of the analysis using a Log2FC of 1.2. Therefore, we decided not to change the original figures. We include the scatter plots with the Log2FC of 1.5 for the reviewer so that they can compare them with those in the first version of the manuscript.

Reviewer Concern 3.

DMSO is used as a vehicle and therefore as a control. I understand that this cannot be changed. But I would like to see a paragraph in the discussion stating that DMSO probably also induced DoGs as this is a major stress for the cells.

Answer:

We agree with the reviewer on this point and in the discussion have added the following in Page 9, line 308:

"In all the experiments analyzed here, DMSO is the vehicle that is used, both with TPL and with THZ1. In all cases, DoGs are detected in cells treated with DMSO. Therefore, it cannot be ruled out that DMSO causes some type of stress that can induce the generation of DoGs."

Reviewer 3 Report

Comments and Suggestions for Authors

The article by Rios et al. investigates how stress from inhibited transcription initiation can lead to overexpression of some genes and the generation of DoGs in breast cancer cells, a response similar to other types of cellular insults such as oxidative stress, heat shock and viral infections. The research also finds that DoGs are not unique to breast cancer cells but also occur in pancreatic cancer cells, indicating a broader relevance. The study concludes that transcription inhibition triggers a typical stress response in cells, leading to both gene overexpression and DoG generation, which could have implications in cancer treatment resistance. The research methodology employed in this study is meticulously designed, and the data presented is both robust and compelling. Based on these strengths, I recommend this manuscript for publication.

Author Response

We would like to thank the reviewers for their comments and for the time dedicated to reviewing this manuscript. The suggested comments have significantly improved this work.